

# Identification of an additive interaction using parameter regularization and model selection in epidemiology

Chanchan Hu[1,*], Zhifeng Lin[1,*], Zhijian Hu[1,2] and Shaowei Lin[1]

[1] Department of Epidemiology and Health Statistics, Fujian Medical University, Fuzhou, Fujian, China
[2] Key Laboratory of Ministry of Education for Gastrointestinal Cancer, Fujian Medical University, Fuzhou, Fujian, China
* These authors contributed equally to this work.

Corresponding authors
Zhijian Hu, huzhijian@fjmu.edu.cn
Shaowei Lin, linsw@fjmu.edu.cn

## ABSTRACT

**Background:** In epidemiology, indicators such as the relative excess risk due to interaction (RERI), attributable proportion (AP), and synergy index (S) are commonly used to assess additive interactions between two variables. However, the results of these indicators are sometimes inconsistent in real world applications and it may be difficult to draw conclusions from them.

**Method:** Based on the relationship between the RERI, AP, and S, we propose a method with consistent results, which are achieved by constraining $e^{\theta_3} - e^{\theta_1} - e^{\theta_2} + 1 = 0$, and the interpretation of the results is simple and clear. We present two pathways to achieve this end: one is to complete the constraint by adding a regular penalty term to the model likelihood function; the other is to use model selection.

**Result:** Using simulated and real data, our proposed methods effectively identified additive interactions and proved to be applicable to real-world data. Simulations were used to evaluate the performance of the methods in scenarios with and without additive interactions. The penalty term converged to 0 with increasing λ, and the final models matched the expected interaction status, demonstrating that regularized estimation could effectively identify additive interactions. Model selection was compared with classical methods (delta and bootstrap) across various scenarios with different interaction strengths, and the additive interactions were closely observed and the results aligned closely with bootstrap results. The coefficients in the model without interaction adhered to a simplifying equation, reinforcing that there was no significant interaction between smoking and alcohol use on oral cancer risk.

**Conclusion:** In summary, the model selection method based on the Hannan-Quinn criterion (HQ) appears to be a competitive alternative to the bootstrap method for identifying additive interactions. Furthermore, when using RERI, AP, and S to assess the additive interaction, the results are more consistent and the results are simple and easy to understand.

## INTRODUCTION

In epidemiology, it is important to determine the interaction between two factors of disease risk, as this information is closely related to disease prevention and appropriate interventions (*Rothman, Greenland & Lash, 2008*; *Diaz-Gallo et al., 2021*). Generally, the interactions are evaluated by including a product interaction term of the two factors in the model. However, epidemiologists frequently employ exponential models, such as logistic regression or Cox regression, to analyze the disease rates and risks, in which the product interaction term is thought to be the multiplicative interaction (*Rothman, Greenland & Lash, 2008*). As Rothman and others have pointed out, it is more meaningful and interesting to evaluate the interaction on an additive scale instead of a multiplicative scale (*Rothman, 1976*; *VanderWeele & Robins, 2007*; *Whitcomb & Naimi, 2023*).

For example, let A and B denote two binary risk factors measured in an epidemiologic study, with their presence and absence reflected by 1 and 0, respectively. $RR_{ij}$ denotes the risk when A = i and B = j. The additive interaction is defined as the difference between the risk difference of A when moving across levels of B and the risk difference of B when moving across levels of A (*Rothman, Greenland & Lash, 2008*). Thus:

$$(RR_{11} - RR_{01}) - (RR_{10} - RR_{00}) = (RR_{11} - RR_{10}) - (RR_{01} - RR_{00}).$$

The additive interaction can be measured by IC $= RR_{11} - RR_{10} - RR_{01} + RR_{00}$, which is known as an interaction contrast. IC $= 0$, if and only if the risk differences for A are constant across B and the risk differences for B are constant across A; that is, $RR_{11} - RR_{01} = RR_{10} - RR_{00}$ or $RR_{11} - RR_{10} = RR_{01} - RR_{00}$, which correspond to additivity, thus there is no additive interaction between factors A and B. Departure from additivity implies the presence of an additive interaction. Specifically, superadditivity (or synergy) is defined as a "positive" departure, which corresponds to IC $> 0$. Subadditivity (or antagonism) is a "negative" departure, which corresponds to IC $< 0$.

*Rothman (1986)* proposed three measures to estimate departure from additivity: the relative excess risk due to interaction (RERI),

$$RERI = RR_{11} - RR_{10} - RR_{01} + 1,$$

the attributable proportion due to interaction (AP),

$$AP = \frac{RERI}{RR_{11}} = \frac{RR_{11} - RR_{10} - RR_{01} + 1}{RR_{11}},$$

and the synergy index (S)

$$S = \frac{RR_{11} - 1}{(RR_{10} - 1) + (RR_{01} - 1)} = \frac{RR_{11} - 1}{RR_{10} + RR_{01} - 2}.$$

If there is no additive interaction, both RERI and AP will be equal to 0, and S will be equal to 1. There are few studies concerning appropriate statistical methods for calculating the confidence interval of these measures (*Hosmer & Lemeshow, 1992*; *Assmann et al., 1996*; *Knol et al., 2007*; *Zou, 2008*; *Richardson & Kaufman, 2009*; *Kuss, Schmidt-Pokrzywniak & Stang, 2010*; *Nie et al., 2010*; *Chu, Nie & Cole, 2011*). However,

results based on the confidence interval of each measure may be inconsistent. For example, the confidence interval of RERI contains a 0, but AP does not. This can cause confusion as to whether the additive interaction is present.

It is noted that $RR_{00} = 1$ and IC can rewritten as:

$$IC = (RR_{11} - RR_{00}) - (RR_{10} - RR_{00}) - (RR_{01} - RR_{00}).$$

Hence, all three measures are based on IC. IC = 0 if and only if RERI = AP = 0 and S = 1. This article describes two methods of using IC directly to identify the additive interaction. The first method is based on regularization (*Bickel et al., 2006*), and the other is based on model selection (*Burnham & Anderson, 2004*). The regularization method includes the regularization term |IC| based on L1-norm in the model to estimate the parameters. By the nature of L1-norm, it may result in parameters shrinkage and variable selection (*Tibshirani, 2011*). Once the tuning parameter is determined IC = 0 can be inferred. The model selection method employs model selection to infer whether IC = 0. We tested two models, with and without IC = 0. Then, the two models were compared using the generalized Akaike information criterion (GAIC) (*Lv & Liu, 2014*) or likelihood ratio test (*Peers, 1971*). There is no additive interaction between the two factors when IC = 0 is favored.

## METHODS

### Estimating the additive interaction using logistic regression

In this method, we let y denote the binary outcome variable labeled by 0 and 1. Z is a vector of the potential confounders which is adjusted in the model and whose dimension, $p$, is dependent on the number of potential confounders.

Using the method proposed by *Rothman (1986)*, the estimates of RERI, AP, and S can be estimated from the output of a multiple logistic regression

$$logit(p) = \log(\frac{p}{1-p}) = \beta_0 + A\beta_1 + B\beta_2 + AB\beta_3 + Z\gamma \tag{1}$$

where $p = P(y = 1|A, B, z)$ is the corresponding probability of outcome variable given the factors A, B and Z, and $\beta_i (i = 0, 1, 2, 3)$ and $\gamma \in R^p$ is the parameter vector corresponding to the two risk factors and confounders respectively. if $\hat{\beta}_i (i = 0, 1, 2, 3)$ are used to stand for the estimated logistic regression coefficients of $\beta_i (i = 0, 1, 2, 3)$, then the RERI, AP, and S can be estimated as

$$\widehat{RERI} = e^{\hat{\beta}_1 + \hat{\beta}_2 + \hat{\beta}_3} - e^{\hat{\beta}_1} - e^{\hat{\beta}_2} + 1$$

$$\widehat{AP} = \frac{e^{\hat{\beta}_1 + \hat{\beta}_2 + \hat{\beta}_3} - e^{\hat{\beta}_1} - e^{\hat{\beta}_2} + 1}{e^{\hat{\beta}_1 + \hat{\beta}_2 + \hat{\beta}_3}}$$

and

$$\widehat{S} = \frac{e^{\hat{\beta}_1 + \hat{\beta}_2 + \hat{\beta}_3} - 1}{e^{\hat{\beta}_1} + e^{\hat{\beta}_2} - 2}.$$

The confidence interval estimator can be calculated based on normal distribution (RERI and AP) or log-normal (S) assumption. To derive the necessary estimator of the variance for these three measures, we used the delta method which based on the first order approximation of a Taylor Series expansion (*Hosmer & Lemeshow, 1992*). The variance and covariance of $\hat{\beta}_i (i = 1, 2, 3)$ are easily found from the logistic regression outcome in most packages.

As *Rothman (1986)* suggested before running the logistic regression, the formula can be simplified by combining the two risk factors into one variable with four levels. Specifically, given the two risk factors, A and B, the new variable is recoded as $A_iB_j(i, j = 0, 1)$, where the subscript indicates the corresponding value of the variable. Then, the new variable is seen as a categorical variable. Using dummy variable encoding and taking the $A_0B_0$ as reference level, the new variable is incorporated into the logistic regression model:

$$logit(p) = \theta_0 + \theta_1 I(A_1B_0) + \theta_2 I(A_0B_1) + \theta_3 I(A_1B_1) + z\gamma \quad (2)$$

where $I(\cdot)$ is the indicator function. Let $\hat{\theta}_i(i = 0, 1, 2, 3)$ denotes the estimated coefficients of $\theta_i(i = 0, 1, 2, 3)$. In Eqs. (1) and (2) both models are saturated and the two estimations are equivalent. Furthermore, it is clear that $\hat{\theta}_1 = \hat{\beta}_1$, $\hat{\theta}_2 = \hat{\beta}_2$ and $\hat{\theta}_3 = \hat{\beta}_1 + \hat{\beta}_2 + \hat{\beta}_3$. In terms of $\hat{\theta}$'s, the RERI, AP, and S can be rewritten as

$$\widehat{RERI} = e^{\hat{\theta}_3} - e^{\hat{\theta}_1} - e^{\hat{\theta}_2} + 1$$

$$\widehat{AP} = \frac{e^{\hat{\theta}_3} - e^{\hat{\theta}_1} - e^{\hat{\theta}_2} + 1}{e^{\hat{\theta}_3}}$$

and

$$\widehat{S} = \frac{e^{\hat{\theta}_3} - 1}{e^{\hat{\theta}_1} + e^{\hat{\theta}_2} - 2},$$

which yield the same point estimates given above. Based on the variances and covariance of $\hat{\theta}_i(i = 1, 2, 3)$. The confidence intervals of these measures are calculated as described above, using the delta method.

When the confidence interval of RERI and AP contain a 0, or of S contains 1, this indicates a lack of the additive interaction between two risk factors. If these values are not present, there is an additive interaction.

## Detect additive interaction based on regularized estimation of logistic regression

To establish notation, consider the logistic regression with sample size $n$, and assume that $\{y_i, A_i, B_i, z_i\}(i = 1, 2, \cdots, n)$ with $n$ observations which have identical but independent distribution. To make estimation and expression simple, Eq. (2) is adopted and rewritten. Let $X_i = (\tilde{X}_i, z_i)$, where $\tilde{X}_i$ is the $i$th row of the design matrix corresponding to the new combined variable, which is composed of zeros and ones denoting the combination of levels for two risk factors A and B. Logistic regression models show the conditional probability $p(X_i) = P(Y = 1|X_i)$ by

$$logit(p(X_i)) = \eta(X_i)$$

with

$$\eta(X_i) = \theta_0 + \tilde{X}_i\theta_{ab} + z_i\gamma = \theta_0 + X_i\tilde{\theta}$$

where $\theta_0$ is the intercept and $\theta_{ab} = (\theta_1, \theta_2, \theta_3)^T \in R^3$, $\tilde{\theta} = (\theta_{ab}^T, \gamma^T)^T \in R^{3+p}$ is the parameter vector corresponding to the risk factors and confounders. For simplicity, let $\theta \in R^{4+p}$ denote the whole parameter vector, *i.e.*, $\theta = (\theta_0, \tilde{\theta}^T)^T$.

The standard maximum likelihood estimation for the logistic regression estimates the coefficients $\theta$ by solving the following equation

$$\hat{\theta} = \arg\min_\theta\{-l(\theta)\} \tag{3}$$

where $l(\cdot)$ is the log-likelihood function:

$$l(\theta) = \sum_{i=1}^n y_i\eta(X_i) - \log[1 + \exp\{\eta(X_i)\}].$$

The LASSO reduced the relative excess risk due to the additive interaction (*Tibshirani, 2011*) and a penalty terms can then be added to Eq. (3):

$$\hat{\theta} = \arg\min_\theta\{-l(\theta)\} \quad s.t. \quad |e^{\theta_3} - e^{\theta_1} - e^{\theta_2} + 1| \leq t \tag{4}$$

where $t \geq 0$ is the tuning parameter which controls the amount of penalization. Solving $\hat{\theta}$ in Eq. (4) is equivalent to the "Lagrange" version of the problem

$$\hat{\theta} = \arg\min_\theta\{-l(\theta) + \lambda|e^{\theta_3} - e^{\theta_1} - e^{\theta_2} + 1|\} \tag{5}$$

where $\lambda \geq 0$. There is a one-to-one correspondence between $t$ and $\lambda$, whose values are regularly chosen by a model selection procedure such as K cross-validation (*Fushiki, 2011*), Akaike information criterion (AIC) or Bayesian information criterion (BIC) (*Burnham & Anderson, 2004*). If $\hat{\theta}(\lambda)$ minimizes Eq. (5), then it also solves Eq. (4) with $t = |e^{\hat{\theta}_3(\lambda)} - e^{\hat{\theta}_1(\lambda)} - e^{\hat{\theta}_2(\lambda)} + 1|$.

It should be noted that $e^{\hat{\theta}_3(\lambda)} - e^{\hat{\theta}_1(\lambda)} - e^{\hat{\theta}_2(\lambda)} + 1 = 0$ when $t$ equals 0 or $\lambda$ is large enough. This means that IC = 0, thus all three additive measures will equal 0 and it indicates the lack of additive interaction. Therefore, if there is no additive interaction, it hopes that $\lambda$ would be large in model selection. When $e^{\hat{\theta}_3(\lambda)} - e^{\hat{\theta}_1(\lambda)} - e^{\hat{\theta}_2(\lambda)} + 1 \neq 0$, the RERI, AP, and S can be calculated and the additive interaction can be inferred that there is an additive scale interaction without requiring additional information, such as the confidence interval.

## Detect additive interaction based on model selection

Although the regularized estimation of logistic regression can determine if there is an additive scale interaction, it depends on the choice of tuning parameter $t$ or $\lambda$. Whatever cross-validation or information criterion, it fixes the value through trial and error. In detail, given a collection of tuning parameter values in advance, for example $\lambda = \{\lambda_1, \cdots, \lambda_l\}$. For each value of $\lambda$, Eq. (5) is solved. After $\lambda = \{\lambda_1, \cdots, \lambda_l\}$ is executed, the best $\lambda^*$ can be determined based on K cross-validation or information criterion.

Model selection is key for determining whether there is an additive interaction (IC = 0). Specifically, two models are fitted; one (model $M_1$) is the logistic regression without any constraint, which is solving the Eq. (3); another (model $M_2$) is the logistic regression with the constraint $e^{\hat{\theta}_3} - e^{\hat{\theta}_1} - e^{\hat{\theta}_2} + 1 = 0$, which is equivalent to solving Eq. (5) with large $\lambda$. In contrast to regularized estimation, it only fits two logistic models.

In regard to model selection, let $l(\hat{\theta})$ be the maximum value of the likelihood function for the model. For a parametric statistical model and within a likelihood-based inferential procedure, the fit of model can be assessed by its fitted global deviance (GDEV), defined as $GDEV = -2l(\hat{\theta})$. $M_1$ and $M_2$ may be evaluated with a fitted global deviance $GDEV_1$ and $GDEV_2$, and degrees of freedom $df_1$ and $df_2$, respectively. The two models, $M_1$ and $M_2$, are nested and may be compared using the likelihood ratio test or the generalized Akaike information criterion (GAIC) (*Lv & Liu, 2014*; *Peers, 1971*). The test statistic of the likelihood ratio is

$$\Lambda = GDEV_2 - GDEV_1$$

which has an asymptotic $\chi^2_d$ distribution under the null hypothesis so the model $M_2$ is recommended, where $d = df_2 - df_1$. The GAIC is obtained by adding a penalty $\kappa$ for each degree of freedom $df$ used in the model to the fitted deviance as follows:

$$GAIC(\kappa) = GDEV + \kappa \times df. \qquad (6)$$

Then the model with the smallest value of the criterion $GAIC(\kappa)$ is selected. AIC and BIC are used most often. $\kappa$ corresponds with $\kappa = 2$ and $\kappa = \ln(n)$ in Eq. (6), respectively:

$$AIC = -2 \cdot l(\hat{\theta}) + 2df,$$

and

$$BIC = -2 \cdot l(\hat{\theta}) + \ln(n)df.$$

The AIC typically leads to overfitting in model selection, while the BIC leads to underfitting (*McLachlan & Peel, 2000*). HQ is another special case of $GAIC(\kappa)$ to consider (*Burnham & Anderson, 2002*),

$$HQ = -2 \cdot l(\hat{\theta}) + 2\ln(\ln(n)) \cdot df.$$

When $n \geq 15$, $\kappa = 2\ln(\ln(n))$ is between 2 and $\ln(n)$, it is considered to be the compromise of AIC and BIC in the penalty terms.

There is only one constraint equation in model $M_2$, so the difference of the degree freedom of the two models equals one. Therefore, when the difference of the GDEV between the two models, $\ln(n)$ and $2\ln(\ln(n))$, is greater than 2 for AIC, BIC, and HQ, respectively, the logistic regression without constraint is chosen and there is an additive interaction between the two risk factors A and B.

## Simulation

We conducted simulation studies whose additive interactions were present or absent based on *a priori* knowledge to assess the performance of detecting the additive interaction based

on regularized estimation of logistic regression or model selection. For simplicity, only two risk factors A and B were considered, regardless of the potential confounders, Z, in all simulation scenarios. The data were generated from the model

$$Y \sim Bernoulli(p)$$

$$logit(p) = \theta_0 + \theta_1 I(A_1 B_0) + \theta_2 I(A_0 B_1) + \theta_3 I(A_1 B_1) \tag{7}$$

where $Bernoulli(\cdot)$ is Bernoulli distribution, and $\theta = (\theta_0, \theta_1, \theta_2, \theta_3)^T \in R^4$ is the coefficients with intercept.

## Regularized estimation

According to *Zou, Hastie & Tibshirani (2007)*, the effective degree of freedom of the LASSO was the number of nonzero coefficients. However, unless the condition $e^{\hat{\theta}_3(\lambda)} - e^{\hat{\theta}_1(\lambda)} - e^{\hat{\theta}_2(\lambda)} + 1 = 0$ was met, the degree of the model *df* would not change. This would make the information criterion fail to determine the $\lambda$. Thus, the value of $\lambda$ was tuned using generalization error which calculated GDEV on new data and can be seen as simplified of the cross-validation (*Golub, Heath & Wahba, 1979*).

Two datasets were generated with $n = 600$ according to Eq. (7), where the coefficients were set to be $\theta = (0, 1, 1, 1.49)^T$ and $\theta = (0, 1, 1, 1.90)^T$, corresponding to models without and with interaction, respectively. The AP is 1/3 in interaction scenario, which can be regarded as weak synergy (*Hosmer & Lemeshow, 1992*). For each dataset, it was split into the training set and testing set with 400 and 200 cases, respectively. The coefficients were estimated using the training set, but the generalization error was calculated on the testing set.

Figure 1 showed the relation between the coefficients and tuning parameter $\lambda$, along with the penalty term $|e^{\hat{\theta}_3(\lambda)} - e^{\hat{\theta}_1(\lambda)} - e^{\hat{\theta}_2(\lambda)} + 1|$. The penalty term $|e^{\hat{\theta}_3(\lambda)} - e^{\hat{\theta}_1(\lambda)} - e^{\hat{\theta}_2(\lambda)} + 1|$ gradually converged to 0 as the tuning parameter $\lambda$ increased. However, the primary concern was whether there was an additive scale interaction rather than the path of the coefficients. Based on the tuning parameter $\lambda$ selected by generalization error, the final models matched the *a priori* information in two scenarios. Specifically, in the scenario in which there is no interaction, the penalty term $e^{\hat{\theta}_3(\lambda)} - e^{\hat{\theta}_1(\lambda)} - e^{\hat{\theta}_2(\lambda)} + 1 = 0$ is met, which indicated there was not additive interaction. While, in the scenario with an interaction, the penalty term $e^{\hat{\theta}_3(\lambda)} - e^{\hat{\theta}_1(\lambda)} - e^{\hat{\theta}_2(\lambda)} + 1 = 0$ disobeyed, which indicated there was additive scale interaction.

## Model selection

In order to assess the performance of detecting additive scale interactions based on model selection, a simulation study was conducted to compare the classical method, such as the delta and bootstrap methods. Model selection used AIC, BIC, and HQ. Both the delta and bootstrap methods employed the three indices: RERI, AP, and S, as denoted by RERI_D, AP_D, S_D, RERI_B, AP_B, and S_B.

For the simulation study, seven scenarios were concerned (Table 1). With the exception of scenarios S2 and S7, all scenarios were derived from the study by *Assmann et al. (1996)*.

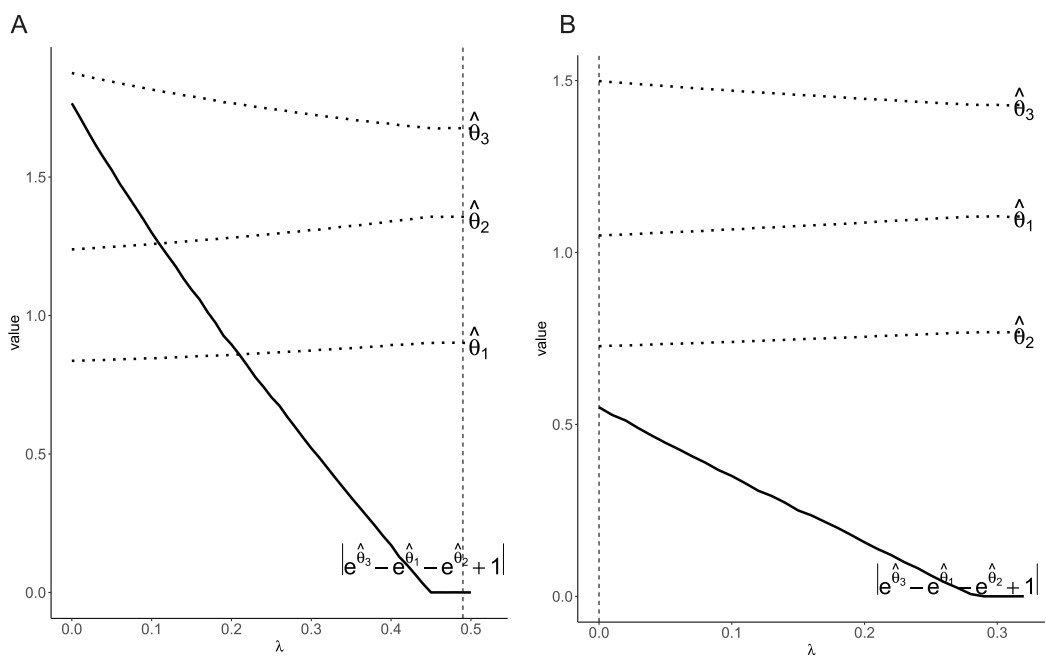

**Figure 1 Plot of the coefficients and penalty term for different values of $\lambda$.** The dashed line indicates the $\lambda$ tuned *via* generalization error. (A) Without interaction; (B) with interaction.

**Table 1 Scenarios setting for simulation.**

| Scenario | $OR_{10}$ | $OR_{01}$ | $OR_{11}$ | RERI | AP | S |
|---|---|---|---|---|---|---|
| S1 | 4 | 5 | 20 | 12 | 0.60 | 2.71 |
| S2 | 4 | 5 | 16 | 8 | 0.50 | 2.14 |
| S3 | 4 | 5 | 12 | 4 | 0.33 | 1.57 |
| S4 | 4 | 5 | 8 | 0 | 0.00 | 1.00 |
| S5 | 4 | 5 | 6 | −2 | −0.33 | 0.71 |
| S6 | 4 | 5 | 4 | −4 | −1.00 | 0.43 |
| S7 | 4 | 5 | 2 | −6 | −3.00 | 0.14 |

Due to RERI = AP = 0 and S = 1 in scenario S4, there was no additive interaction. The model interaction occurred in all other models. Thus, those scenarios determined which models had strong synergy, weak synergy, no interaction, weak antagonism, or strong antagonism.

In each scenario, 1,000 samples were generated based on Eq. (7) with sample sizes 400, 600 and 1,000, respectively. The parameters $\theta$ were the logarithm of the ORs. For example, in scenario S1, parameters $\theta$ were set in Eq. (7) as

$$\theta = [0, \log(4), \log(5), \log(20)]^T = (0, 1.386, 1.609, 2.996)^T.$$

For each sample, the interaction detected model selection with $GAIC(\kappa)$ and the confidence intervals of RERI, AP, and S with delta and bootstrap methods. Then the methods were compared based on the percentage of times with no additive interaction.

**Table 2 The percentage of no interaction with 1,000 samples.**

| Sizes | Scenario | AIC | HQ | BIC | RERI_D | AP_D | S_D | RERI_B | AP_B | S_B |
|-------|----------|-----|-----|-----|--------|------|-----|--------|------|-----|
| 400 | S1 | 0.076 | 0.168 | 0.345 | 0.614 | 0.100 | 0.207 | 0.185 | 0.184 | 0.185 |
| | S2 | 0.161 | 0.299 | 0.503 | 0.762 | 0.199 | 0.345 | 0.302 | 0.302 | 0.302 |
| | S3 | 0.378 | 0.547 | 0.747 | 0.937 | 0.434 | 0.591 | 0.557 | 0.556 | 0.556 |
| | S4 | 0.790 | 0.919 | 0.975 | 0.997 | 0.873 | 0.930 | 0.911 | 0.910 | 0.910 |
| | S4* | 0.801 | 0.914 | 0.976 | 0.995 | 0.873 | 0.932 | 0.919 | 0.918 | 0.918 |
| | S5 | 0.686 | 0.838 | 0.926 | 0.943 | 0.950 | 0.844 | 0.845 | 0.846 | 0.846 |
| | S6 | 0.044 | 0.104 | 0.248 | 0.272 | 0.374 | 0.118 | 0.117 | 0.119 | 0.119 |
| | S7 | 0.000 | 0.000 | 0.000 | 0.000 | 0.000 | 0.001 | 0.000 | 0.000 | 0.000 |
| 600 | S1 | 0.011 | 0.051 | 0.151 | 0.147 | 0.025 | 0.055 | 0.053 | 0.053 | 0.053 |
| | S2 | 0.054 | 0.128 | 0.319 | 0.321 | 0.080 | 0.145 | 0.135 | 0.136 | 0.136 |
| | S3 | 0.231 | 0.411 | 0.636 | 0.664 | 0.312 | 0.433 | 0.413 | 0.412 | 0.412 |
| | S4 | 0.787 | 0.918 | 0.975 | 0.980 | 0.873 | 0.924 | 0.911 | 0.911 | 0.910 |
| | S4* | 0.789 | 0.913 | 0.975 | 0.978 | 0.868 | 0.924 | 0.910 | 0.910 | 0.910 |
| | S5 | 0.621 | 0.811 | 0.921 | 0.862 | 0.916 | 0.814 | 0.809 | 0.808 | 0.808 |
| | S6 | 0.008 | 0.020 | 0.081 | 0.041 | 0.080 | 0.023 | 0.024 | 0.024 | 0.024 |
| | S7 | 0.000 | 0.000 | 0.000 | 0.000 | 0.000 | 0.000 | 0.000 | 0.000 | 0.000 |
| 1,000 | S1 | 0.001 | 0.002 | 0.012 | 0.003 | 0.001 | 0.002 | 0.001 | 0.001 | 0.001 |
| | S2 | 0.004 | 0.012 | 0.095 | 0.040 | 0.009 | 0.013 | 0.015 | 0.015 | 0.015 |
| | S3 | 0.087 | 0.211 | 0.442 | 0.324 | 0.147 | 0.215 | 0.198 | 0.198 | 0.198 |
| | S4 | 0.742 | 0.895 | 0.981 | 0.950 | 0.834 | 0.906 | 0.884 | 0.888 | 0.887 |
| | S4* | 0.766 | 0.902 | 0.975 | 0.946 | 0.853 | 0.904 | 0.892 | 0.892 | 0.892 |
| | S5 | 0.512 | 0.724 | 0.894 | 0.760 | 0.835 | 0.714 | 0.707 | 0.707 | 0.707 |
| | S6 | 0.000 | 0.000 | 0.007 | 0.000 | 0.001 | 0.000 | 0.000 | 0.000 | 0.000 |
| | S7 | 0.000 | 0.000 | 0.000 | 0.000 | 0.000 | 0.000 | 0.000 | 0.000 | 0.000 |

**Note:**
* The number of simulations is 10,000.

For the bootstrap method (*Efron, 1982*), 300 bootstrap samples were tested separately within $Y = 0$ and $Y = 1$. Therefore, the number of $Y = 0$ and $Y = 1$ in the bootstrap samples were identical to the original sample.

The simulation results for models with no interaction are summarized in Table 2. The percentage was calculated as the number of times that model correctly identified in scenario S4 (without interaction), but it was the rate of times that model misidentified the other scenarios (with interaction). Thus, the correct model was chosen as that which was one minus the error rate in scenarios with interaction. Scenario S4 provided an assessment of the Type I error rate when testing for additive interactions using different methods, while the other scenarios provided an evaluation of the power of these methods when testing for additive interactions with varying effect sizes. Clearly, among all methods, the percentage became smaller and smaller as the sample size increased and/or the interaction effect was augmented. This was due to the fact that it was more likely to get significant results the larger the sample size and/or the effect size (*Papoulis, 1990*) in favor of the

model with interaction. Additionally, when the interaction effect was weak it was due to a lack of power to detect the interaction.

The bootstrap method is recommended for use in various applied fields (*Jalali et al., 2022*; *Miwakeichi & Galka, 2023*; *Kaity et al., 2023*) as well as for interaction detection (*Assmann et al., 1996*). Whatever measures were used, the percentages of no interaction were very close in various scenarios. Therefore, the bootstrap method was used as a benchmark. In contrast, however, there was a lot of variation from the delta method, especially when the sample size was relatively small. For example, in scenario S2 for sample size of 400, the percentages of no interaction corresponding to RERI_D, AP_D, and S_D were 0.762, 0.199, and 0.345, respectively, and its standard deviation was 0.292. Using bootstrap, the percentages of no interaction corresponding to RERI_B, AP_B. and S_B were 0.302, the standard deviation almost equaled to 0. This indicated that there was often conflict among the results in detecting interaction *via* RERI_D, AP_D, and S_D. However, based on model selection, due to $e^{\hat{\theta}_3} - e^{\hat{\theta}_1} - e^{\hat{\theta}_2} + 1 = 0$, there were consistent results regardless of which measure used.

Using the bootstrap method as the criterion, the S_D had the smallest deviation and RERI_D had the largest deviation in almost all scenarios. Specifically, S_D was close to S_B, but RERI and AP were not. In contrast to the bootstrap method, RERI_D was favored without interaction, regardless of whether it was synergy or antagonism interaction. However, AP_D favored the model with a synergy interaction and without an antagonistic interaction.

Obviously, when $\kappa$ increased, the $GAIC(\kappa)$ also increased and tended to favor the model without interaction. Since $\kappa = 2, 2\ln(\ln(n))$ and $\ln(n)$ for AIC, HQ, and BIC, respectively, for each scenario. The percentage of no interactions were rising, as shown in Table 2. It is noted that the results of HQ were almost the same as the results of the bootstrap method. The difference between the results of HQ and the bootstrap method were less than 2% in all scenarios. In contrast, there was a smaller percentage of no interaction for AIC, but larger for BIC. This meant that the AIC tended to overestimate the interaction, while the BIC tended to underestimate the interaction (*McLachlan & Peel, 2000*).

## Real data

The data came from a case-control study of oral cancer, kindly supplied by *Rothman & Keller (1972)*. The two factors were smoking and alcohol use. There were a total of 458 participants who were male veterans under the age of 60s. The study investigated the effects of smoking and alcohol use on oral cancer. The distribution of data is shown in Table 3. It was used to illustrate their methods on how to calculate the confidence interval for measures of interaction by several authors (*Hosmer & Lemeshow, 1992*; *Zou, 2008*; *Richardson & Kaufman, 2009*; *Chu, Nie & Cole, 2011*). The confidence intervals were different among their methods. However, their results consistently showed that there was not an additive scale interaction.

Using the interaction detection based on model selection, two logistic models were employed. $M_u$ and $M_c$ indicated which model had the constraint $e^{\hat{\theta}_3} - e^{\hat{\theta}_1} - e^{\hat{\theta}_2} + 1 = 0$. The result is summarized in Table 4. The GDEV of model $M_u$ and $M_c$ were 605.930 and

**Table 3 Distribution of exposures among cases and controls in the oral cancer example[a].**

|          | Neither | Smoking only | Alcohol only | Smoking and alcohol |
|----------|---------|--------------|--------------|---------------------|
| Cases    | 3       | 8            | 6            | 225                 |
| Controls | 20      | 18           | 12           | 166                 |

**Note:**
   [a] Data was from the example presented by *Rothman & Keller (1972)*.

**Table 4 The coefficients and GDEV of the oral cancer example in logistic regression.**

| Model   | $\hat{\theta}_1$ | $\hat{\theta}_2$ | $\hat{\theta}_3$ | GDEV    |
|---------|--------|--------|--------|---------|
| $M_u$   | 1.204  | 1.086  | 2.201  | 605.927 |
| $M_c$   | 1.882  | 1.497  | 2.306  | 607.730 |

607.727, respectively. Thus, the difference between them was 1.803, which was less than two. The model without an interaction was favored, regardless of which criterion was used (AIC, BIC, or HQ). Therefore, consensus had to be reached with other authors. Remarkably, the coefficients of model $M_c$ without an additive interaction met the following equation:

$$IC = e^{2.306} - e^{1.882} - e^{1.497} + 1 = 0,$$

which would make the interpretation of the result simple. Additionally, we validated our algorithm in two other cases: a hypertension case (*Zou, 2008*) and a congenital heart disease case (*Nie et al., 2016*); the results were consistent with the authors' analysis (Tables S1–S4).

## DISCUSSION

It is vital to recognize that disease prevention and intervention are closely related to its influencing factors and their interactions. In epidemiology, additive interaction plays an important role (*Rothman, Greenland & Lash, 2008*). There have been many discussions concerning appropriate methods for constructing confidence intervals or credible intervals (CIs) for RERI, AP, and S (*Hosmer & Lemeshow, 1992*; *Assmann et al., 1996*; *Kuss, Schmidt-Pokrzywniak & Stang, 2010*). If the CIs do not contain the null value (0 with RERI and AP, 1 with S), then the identification of the additive interaction is confirmed. In addition to CIs, this article proposes a novel method to assess the additive interaction. This method works because it is concerned with whether the constraint $e^{\hat{\theta}_3} - e^{\hat{\theta}_1} - e^{\hat{\theta}_2} + 1 = 0$ is met. When the constraint is met, this means the RERI and AP equals 0 and S equals 1, so there is no additive interaction between the two factors. Otherwise, the additive interaction exists.

In this article, there are two methods to verify that the constraint is met, using regularized estimation or model selection. However, it is easy to see that, when using the same information criterion in choosing the tuning parameter $\lambda$ and model selection, the conclusion would be the same, regardless of the method used. For example, with BIC, if the constraint $e^{\hat{\theta}_3} - e^{\hat{\theta}_1} - e^{\hat{\theta}_2} + 1 = 0$ is met in regularized estimation, the $GAIC(\kappa)$ would be

in agreement with that of the model with constraint, so that the model with the constraint would be favor, *i.e.*, there is no additive interaction between the two factors regardless of which method used, and *vice versa*. As the choice of tuning parameter $\lambda$ is computationally expensive, the model selection method is recommended.

From the simulation studies, the percentage of misidentification decreased as the sample size increased and/or the additive interaction effect was augmented among all methods. Based on the confidence interval method, the performance of the delta method was worse than that of bootstrap method; this effect has been well-documented (*Assmann et al., 1996*). The values of RERI_B, AP_B, and S_B were almost the same in each scenario, but RERI_D, AP_D, and S_D tended to be unusually volatile. Thus, the conclusion based on the delta method with different measures would be inconsistent, but there may be little impact on the bootstrap method. Based on this study, it is a surprise that the performance of S_D was like that of the bootstrap method, although the measurement indicator S was not proposed by a few authors. This might be attributable to confidence interval using the logarithmic function, which may can help the distribution of the S be more normal and stabilize the variance (*Muhsam, 1946*). Based on model selection methods, the percentages of misidentification of HQ were almost the same as that of the bootstrap method. However, AIC tended to overestimate the interaction, while the BIC tended to underestimate the interaction. Thus, HQ is a better choice for model selection.

Furthermore, when employing the confidence interval to assess the additive interaction, no matter what method is used, the result may be inconsistent among different measure indicators. However, using the model selection method proposed by this article, if $\text{RERI} = e^{\hat{\theta}_3} - e^{\hat{\theta}_1} - e^{\hat{\theta}_2} + 1 = 0$, then AP = 0 and S = 1, and vice versa. Therefore, there is always a consistent result, regardless of the different measure indicators. Furthermore, the model selection method can make the interpretation of the result simple and straightforward. Specifically, when there is no additive interaction, using the confidence interval method, even the results of RERI, AP, and S are consistent. Their intervals contain the null values but not equal. In contrast, unlike the confidence interval which involves uncertainty, there is no additive interaction if and only if the interaction contrast equals 0 in the model selection method. For example, in the example of smoking and alcohol, there was not an additive interaction, thus the interaction contrast equals 0, *i.e.*, $e^{2.306} - e^{1.882} - e^{1.497} + 1 = 0$.

In summary, a model selection method based on HQ appears to be a competitive alternative to the bootstrap method for identifying an additive interaction. Furthermore, when using RERI, AP, and S measures to assess the additive interaction, the result is a consistent conclusion with a clear interpretation of the results, which may be useful in practice.

### Funding
The authors received no funding for this work.

## Competing Interests

The authors declare that they have no competing interests.

## Author Contributions

- Chanchan Hu conceived and designed the experiments, performed the experiments, analyzed the data, prepared figures and/or tables, and approved the final draft.
- Zhifeng Lin conceived and designed the experiments, performed the experiments, analyzed the data, prepared figures and/or tables, and approved the final draft.
- Zhijian Hu conceived and designed the experiments, authored or reviewed drafts of the article, and approved the final draft.
- Shaowei Lin conceived and designed the experiments, authored or reviewed drafts of the article, and approved the final draft.

## Data Availability

The raw measurements are available in the Supplemental File.

## Supplemental Information

Supplemental information for this article can be found online at http://dx.doi.org/10.7717/peerj.18304#supplemental-information.

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
