# Peer review of "Identification of an additive interaction using parameter regularization and model selection in epidemiology"

_PeerJ, doi:10.7717/peerj.18304_

## Round 0.1 · original submission · Major Revisions

In this study, the authors aimed to propose a model and validated it based on a study reporting 458 observations wherein they evaluated the interaction of smoking and alcohol use on oral cancer. The main idea of the study is to add a penalty term to the usual log-likelihood function of a logistic regression to constrain deviation from an additive relationship. This study was found well-written and innovative by the reviewers. However, the paper needs additional work, and some major changes are recommended before considering the paper for publication.

Reviewer 1 ·

Basic reporting

This paper proposes two methods for identifying additive interaction in epidemiology: parameter regularization and model selection. The main idea is to add a penalty term to the usual log-likelihood function of a logistic regression to constrain deviation from an additive relationship. This approach is innovative, and while the paper is generally well-written, there is room for further refinement.
1. The authors can treat their methods as a hypothesis test. In their simulation study, they should examine the statistical properties of their new ‘tests’ and compare the results with existing methods for testing additive interaction. The statistical properties to be examined should include type I error rates when the null hypothesis of no additive interaction is true, and power curves when the alternative hypothesis of varying degrees of departure from additive interaction is true. The number of simulations should be increased: for type I error rates, at least 10,000; for power curves, at least 1,000.
2. The authors have used AIC, HQ, and BIC criteria in their paper. However, what about using the plain likelihood ratio test? For one degree of freedom, this would involve using a critical value of 3.84 to determine significance. One more comment: The first time HQ appears in the abstract, it is not spelled out in full.
3. Regarding the tuning parameter for the penalty term, lambda: When lambda is zero, the model is a regular logistic regression with four free theta parameters (theta_0, theta_1, theta_2, theta_3). When lambda tends to infinity, the model becomes a constrained logistic regression with only three free theta parameters, as they are constrained to comply with the additive relationship. So, why don’t the authors approach the problem this way? That is, compare the model fits of the unconstrained ordinary logistic regression and the constrained logistic regression, and dispense with all the developments of parameter regularization and model selection.

Experimental design

same as the above

Validity of the findings

same as the above

Additional comments

same as the above

Reviewer 2 ·

Basic reporting

The manuscript titled "Identification of additive interaction using parameter regularization and model selection in epidemiology" is reasonably well written and informative. The content presented is very clear and easy to understand.

Experimental design

The authors have proposed the model and validated it based on a study reporting 458 observations wherein they evaluated the interaction of smoking and alcohol use on oral cancer. Why haven't the authors validated the proposed method with multiple cases? Why haven't the authors chosen other studies with less number of cases and more number of cases to better validate the model? The reviewer suggests that the authors should thoroughly work on the proposed model with varying number of multiple instances to effectively validate it.

Validity of the findings

As mentioned above, the model should be thoroughly validated.

Additional comments

The authors have cited multiple references but majority of the references are pretty old. Why haven't the authors scanned latest literature to support or validate their findings.

---

## Round 0.2 · accepted · Accept

I would like to thank the authors for the revision of the manuscript. The authors have addressed all comments of the reviewers. The manuscript is now ready for publication.

Reviewer 1 ·

Basic reporting

The authors have adequately addressed my comments.

Experimental design

The authors have adequately addressed my comments.

Validity of the findings

The authors have adequately addressed my comments.

Additional comments

The authors have adequately addressed my comments.

Reviewer 2 ·

Basic reporting

The manuscript titled "Identification of additive interaction using parameter regularization and model selection in epidemiology" is much improved after thorough revision by the authors. The content presented is very clear and easy to understand.

Experimental design

During the revision process, the authors have effectively validated the proposed model with multiple instances which is clearly reflected in the number of tables and supplementary data that is added.

Validity of the findings

As mentioned above, the model was well validated.